Mitochondrial genome of the nonphotosynthetic mycoheterotrophic plant Hypopitys monotropa, its structure, gene expression and RNA editing

Shtratnikova Viktoria Yu 1
http://orcid.org/0000-0001-8804-7281 Schelkunov Mikhail I. 2 3
Penin Aleksey A. 3
Logacheva Maria D. 2 maria.log@gmail.com
1 A. N. Belozersky Institute of Physico-Chemical Biology, Lomonosov Moscow State University , Moscow , Russia
2 Skolkovo Institute of Science and Technology , Moscow , Russia
3 Laboratory of Plant Genomics, Institute for Information Transmission Problems of the Russian Academy of Sciences , Moscow , Russia
Ray David
Electronic publication date: 2020 Jun 19
Publication date: 2020
Volume: 8
Electronic Location ID: e9309
Received 2019 Jun 13; Accepted 2020 May 17
Copyright: © 2020 Shtratnikova et al.
Copyright year: 2020
Copyright holder: Shtratnikova et al.
License: This is an open access article distributed under the terms of the Creative Commons Attribution License, which permits unrestricted use, distribution, reproduction and adaptation in any medium and for any purpose provided that it is properly attributed. For attribution, the original author(s), title, publication source (PeerJ) and either DOI or URL of the article must be cited.
License URL: https://creativecommons.org/licenses/by/4.0/

Keywords: Mitochondrial genome, Mycoheterotrophic plants, Hypopitys monotropa, RNA editing, Non-photosynthetic plants

Funding: Russian Science Foundation project #17-14-01315 Budgetary subsidy to IITP RAS project # 0053-2019-0005 This work was supported by the Russian Science Foundation (project #17-14-01315, RNA editing and gene expression analysis) and budgetary subsidy to IITP RAS (project # 0053-2019-0005, genome analysis). The funders had no role in study design, data collection and analysis, decision to publish, or preparation of the manuscript.

==============================
Heterotrophic plants—plants that have lost the ability to photosynthesize—are characterized by a number of changes at all levels of organization. Heterotrophic plants are divided into two large categories—parasitic and mycoheterotrophic (MHT). The question of to what extent such changes are similar in these two categories is still open. The plastid genomes of nonphotosynthetic plants are well characterized, and they exhibit similar patterns of reduction in the two groups. In contrast, little is known about the mitochondrial genomes of MHT plants. We report the structure of the mitochondrial genome of Hypopitys monotropa, a MHT member of Ericaceae, and the expression of its genes. In contrast to its highly reduced plastid genome, the mitochondrial genome of H. monotropa is larger than that of its photosynthetic relative Vaccinium macrocarpon, and its complete size is ~810 Kb. We observed an unusually long repeat-rich structure of the genome that suggests the existence of linear fragments. Despite this unique feature, the gene content of the H. monotropa mitogenome is typical of flowering plants. No acceleration of substitution rates is observed in mitochondrial genes, in contrast to previous observations in parasitic non-photosynthetic plants. Transcriptome sequencing revealed the trans-splicing of several genes and RNA editing in 33 of 38 genes. Notably, we did not find any traces of horizontal gene transfer from fungi, in contrast to plant parasites, which extensively integrate genetic material from their hosts.

Introduction

Hypopitys monotropa (Ericacae, Monotropoideae) is a nonphotosynthetic plant that obtains carbon from fungi in ectomycorrhizal relationships with tree roots (Bjorkman, 1960). In contrast to most other mycoheterotrophic (MHT) plants, which are very rare and/or very narrowly distributed, Monotropoideae, including H. monotropa, are quite widespread and are associated with old-growth conifer forests. Thus, H. monotropa is used as a model system in studies of plant-mycorrhizal associations and the developmental biology of MHT plants (Olson, 1993, 1990). Recent advances in DNA sequencing have expanded research on mycoheterotrophs into genomics. To date, most attention has been focused on the plastid genomes of MHT plants, which are highly reduced in size and gene content, including a complete absence or pseudogenization of genes of the photosynthesis electron transport chain (for a review, see Graham, Lam & Merckx, 2017). Thus, the MHT lifestyle strongly affects plastids, but what about the mitochondrial genome?

In contrast to animals, in which mitochondrial genomes are usually conserved in size and gene content across large taxonomic groups, in plants, these genomes are highly variable and may be very dissimilar even in closely related species. The size of the angiosperm mitogenome ranges from 66 Kb in the hemiparasitic mistletoe Viscum scurruloideum (Skippington et al., 2015) and 222 Kb in the autotrophic Brassica napus (Handa, 2003) to more than 11 Mb in Silene noctiflora. Despite such huge variations in size, a large fraction of mitochondrial genes—those that encode the components of the oxidative phosphorylation chain complexes and proteins involved in the biogenesis of these complexes—are stable in content and show very low sequence divergence. More variation exists in the group of genes involved in translation, that is, ribosomal proteins and transfer RNAs (Adams et al., 2002; Gualberto et al., 2014), presumably due to the transfer to the nuclear genome that occurred in several plant lineages or the non-essentiality of these genes. In addition, many plant mitochondrial genomes carry open reading frames (ORFs) that potentially encode functional proteins (Qiu et al., 2014); such ORFs are highly lineage specific. Nonphotosynthetic plants are divided into two large groups—those that are parasitic on other plants and those that are mycoheterotropic. To date, only a few complete mitochondrial genomes of nonphotosynthetic plants have been characterized, and most of them belong to parasitic species. A comparative analysis of the mitogenomes of several parasitic, hemiparasitic and autotrophic Orobanchaceae (Fan et al., 2016) showed that the gene content does not depend on trophic specialization in the family range. The mitogenomes of two non-related lineages of parasitic plants, Rafflesiaceae and Cynomoriaceae, also do not show a reduction in gene content, and they provide an example of massive HGT from other plants (including but not limited to their hosts). This is not, however, a trait that is unique to parasitic plants—for example, in Amborella trichopoda, an autotrophic plant from the basal angiosperms, a large fraction of the mitochondrial genome was acquired from green algae, mosses, and other angiosperms (Rice et al., 2013). In contrast, in the hemiparasitic plant V. scurruloideum, the mitogenome is drastically reduced in size and gene content, and it lacks all nine nad genes and matR (Skippington et al., 2015). The mitogenomes of other Viscum species are not reduced in length but exhibit a reduced gene content, similar to that of V. scurruloideum (Petersen et al., 2015; but see Skippington et al., 2017). The available sampling is obviously insufficient even for parasitic plants; regarding mycoheterotrophs, the data are almost completely lacking. Only two mitochondrial genomes of MHT plants have been characterized by date—those of the orchid Gastrodia elata (Yuan et al., 2018) and Epirixanthes elongata (Polygalaceae) (Petersen et al., 2019). The former was characterized as a part of a genome sequencing project and is not completely assembled, being represented by 19 contigs. The G. elata mitogenome is large (~1.3 Mb), while that of E. elongata is much smaller (~0,4 Mb). However, both of these studies lack a comparative analysis with autotrophic relatives that would allow to highlight the changes associated with mycoheterotrophy.

In this context, we set the following objectives: (1) to characterize the structure and gene content of the mitochondrial genome of H. monotropa, (2) to estimate the possibility of horizontal gene transfer (HGT) from fungi, and (3) to study mitochondrial gene expression and RNA editing.

The photosynthetic plant with a characterized mitochondrial genome phylogenetically closest to H. monotropa is cranberry, V. macrocarpon (Fajardo et al., 2014). In this study, we used V. macrocarpon as a basis for comparative analysis aimed at identifying the patterns (if any) of mitochondrial genome changes associated with mycoheterotrophy.

Materials and Methods

Sample collection and sequencing

The procedures for sample collection, DNA and RNA library preparation for most datasets (DNA shotgun and mate-pair libraries and an inflorescence transcriptome library) and sequencing were described in the reports of Logacheva et al. (2016) and Schelkunov, Penin & Logacheva (2018). In addition to the data generated in previous studies, we used four new datasets that represent the transcriptomes of the anthers and ovules of H. monotropa. The samples were collected in the same location as previous samples but in 2018. H. monotropa is not an endangered or threatened plant species; thus, no specific permissions were required for its collection. RNA was extracted using an RNEasy kit (Qiagen, Hilden, Germany) with the addition of the Plant RNA Isolation Aid reagent (Thermo Fisher, Waltham, MA, USA) to the lysis buffer. The removal of ribosomal RNA was performed using a Ribo-Zero plant leaf kit (Illumina, San Diego, CA, USA), and further sample preparation was performed using a NEBNext RNA library preparation kit (New England Biolabs, Ipswich, MA, USA). The libraries were sequenced on the HiSeq 4000 platform (Illumina, San Diego, CA, USA) in 150-nt paired-end mode at the Skoltech Genomics Core Facility. The reads were deposited in the NCBI Sequence Read Archive under BioProject PRJNA573526.

Assembly

Read trimming and assembly were performed as described in the report of Logacheva et al. (2016). The contig coverage was determined by mapping all reads in CLC Genomics Workbench v. 7.5.1 (https://www.qiagenbioinformatics.com/products/clc-genomics-workbench/), requiring at least 80% of a read’s length to align with at least 98% sequence similarity. To find contigs corresponding to the mitochondrial genome, we performed BLASTN and TBLASTX alignment of the V. macrocarpon genome (GenBank accession NC_023338) against all contigs in the assembly. BLASTN and TBLASTX from the BLAST 2.3.0+ suite (Camacho et al., 2009) were used, and the alignment was performed with a maximum allowed e-value of 10−5. Low-complexity sequence filters were switched off to avoid missing genes with an extremely high or low GC content. A contig that corresponded to the plastid genome of H. monotropa also aligned to the mitochondrial genome of V. macrocarpon because of the presence of inserts from the plastid genome in the mitochondrial genome. The complete sequence of the H. monotropa plastome is known from our previous study (Logacheva et al., 2016) and was excluded from further analyses. Contigs showing coverage of less than 10 or those that exhibited matches only to non-mitochondrial sequences when aligned to the NCBI nt database by BLASTN (e-value 10−3) were considered nuclear.

After the procedures described above, seven contigs remained, with lengths of 255, 5,651, 20,627, 102,868, 106,298, 234,240 and 309,413 bp and coverages of 37, 89, 93, 110, 96, 107 and 108 bp, respectively. The lower coverage of the smallest contig was presumably an artifact caused by its length. To understand their order in the mitochondrial genome, we mapped the mate-pair reads from the library with the largest insert size (8,705 ± 2,537 bp) to the contigs and investigated the mate-pair links. The mapping was performed by using CLC Genomics Workbench and 100% of a read’s length was required to align to a contig with a sequence similarity of 100%, to minimize the amount of incorrectly mapped reads. Mate-pair links were visualized by using Circos 0.67–5 (Krzywinski et al., 2009) and investigated manually. In one contig, the mate-pair links spanned the entire length from the end to the start; thus, the contig corresponded to a circular chromosome. The other six contigs were connected into a linear sequence, whose structure is described in the Results section. Gaps between the contigs were closed by GapFiller (Boetzer & Pirovano, 2012), which was run with the following parameters: -m 30 -o 2 -r 0.8 -n 50 -d 100000 -t 100 -g 0 -T 4 -i 100000. The source code of GapFiller was rewritten to allow it to use Bowtie2 as the read mapper. Additionally, we tried to extend the edges of the linear chromosome by using GapFiller to check that there were no assembly forks (sites with two alternative extensions). The absence of such forks indicates that the linearity of the sequence is not a consequence of an assembly problem. Instead, during extension, the coverage gradually decreased to zero, implying a linear chromosome with lengths varying between different copies of the chromosome. To check whether the two chromosomes were assembled correctly, we mapped the reads from all three sequencing libraries by using CLC Genomics Workbench v. 7.5.1, requiring at least 90% of a read’s length to align with at least 99% sequence similarity and checked that the coverage was approximately uniform along the whole length of the chromosomes. As observed during the analysis by GapFiller, we found that the coverage gradually decreased at the edges of the linear chromosome, indicating that different copies of the chromosome in a plant exhibit different lengths of the terminal regions. For the sake of precision, we elongated the terminal regions such that the coverage dropped to 0; thus, the length of the linear chromosome approximately corresponded to the maximal possible length among all copies of the chromosome in the sequenced plant. Additionally, we mapped the reads from the mate-pair library with the largest insert sizes, requiring 100% of a read’s length to align with a sequence similarity of 100%, and we investigated the distribution of mean insert sizes over each genomic position and the number of mate pairs than spanned each position. These values were approximately uniform along both chromosomes (Fig. S1), suggesting that the assembly was correct.

Mitogenome annotation

Initial annotation was performed using Mitofy (Alverson et al., 2010) with further manual correction. To verify exon boundaries, we mapped RNA-seq reads to the sequences of the chromosomes by using CLC Genomics Workbench v. 7.5.1, requiring at least 80% of a read’s length to map with at least 90% sequence similarity. The relaxed setting for the percentage of read length to be mapped allows the mapping reads to span splice junctions, and the relaxed setting for sequence similarity allows the mapping of reads to regions with a high density of RNA editing sites. The alignment was visualized in CLC Genomics Workbench 7.5.1.

To check if there were any protein-coding genes that were missed, we visually inspected the alignment to search for unannotated regions in the chromosomes with high coverage by RNA-seq reads. This search did not add new genes.

The search for genes encoding selenocysteine tRNAs (tRNA-Sec) was performed for H. monotropa and V. macrocarpon by using an online version of Secmarker 0.4 (Santesmasses, Mariotti & Guigó, 2017) with the default parameters. SECIS element prediction was performed by using an online version of SECISEARCH3 (Mariotti et al., 2013) that was current as of 5 May 2017 with the default parameters. Assembled and annotated sequences of H. monotropa mitochondrial genome are deposited in the NCBI Genbank under accession numbers MK990822 and MK990823.

Gene Ontology (GO) terms were assigned to the genes of the mitochondrial genome by the NetGO 2.0 server (You et al., 2019). Genes with their GO terms are provided in Table S2. GO terms with prediction scores <= 0.6 are considered of low confidence in the manual of NetGO and were therefore discarded.

RNA editing analysis

To obtain information on RNA editing sites, we artificially spliced gene sequences and mapped RNA-seq reads to the sequences with the parameters indicated above for RNA-seq read mapping. Variant calling was performed by using CLC Genomics Workbench 7.5.1. We considered a site in a CDS to be edited if it was covered by at least 20 reads and at least 10% of the reads in the corresponding positions differed from the genomic sequence.

To estimate the fraction of RNA editing sites that change the amino acid sequences of the corresponding proteins, we used a measure that has been used in several previous studies (Chen, 2013; Xu & Zhang, 2014; Zhu et al., 2014) but has not yet been given a name. Here, we refer to this measure as dEN/dES. It is calculated similarly to dN/dS, but while dN/dS represents the ratio of nonsynonymous and synonymous substitutions per nonsynonymous and synonymous sites, dEN/dES (“E” stands for “editing” here) represents the ratio of nonsynonymous and synonymous RNA editing events per nonsynonymous and synonymous sites.

To compare the RNA editing sites in the mitochondrial mRNAs of H. monotropa with those in the mitochondrial mRNAs of other species, we used data on RNA editing in Arabidopsis thaliana (Giege & Brennicke, 1999) and Oryza sativa (Notsu et al., 2002). These data are incorporated in the annotations of A. thaliana and O. sativa in GenBank, with accession numbers NC_037304.1 and BA000029.3, respectively.

Search for sequences transferred to the mitogenome

To identify sequences transferred from the plastome to the mitogenome (known as “MIPTs”, which stands for MItochondrial Plastid Transfers), we aligned the plastid genes of Camellia sinensis to the sequences of the H. monotropa mitogenome. Camellia sinensis was chosen because its plastome contains a complete set of typical plastid genes and is phylogenetically closest to H. monotropa (as of May 2017) among all sequenced plants with such a complete set. We searched for transferred genes and not intergenic regions because we were interested in transfers that may have initially been functional. The proteins of Camellia were aligned to the sequences of the mitochondrial chromosomes by using TBLASTN with a maximum allowed e-value of 10−3. tRNA and rRNA coding genes were similarly aligned to the sequences of the chromosomes by using BLASTN. The matching regions of the chromosomes were then aligned by BLASTX to the NCBI nr database (for regions that matched Camellia proteins) and by BLASTN to the NCBI nt database (for regions that matched Camellia RNA coding genes). If the best matches in the database for a region in the mitogenome were to sequences belonging to plastomes, that region was considered a MIPT. To calculate the number of frameshift-inducing indels and nonsense mutations in the transferred regions, these regions were considered together with their 200 bp-long flanking sequences on both ends and aligned to homologous genes from C. sinensis. The alignment was performed by using BLASTN with the default parameters. The resultant alignments were inspected by eye.

To search for possible horizontal gene transfers from fungi, the mitogenome sequences were split into windows of 500 bps each, with a step size of 50 bps. These windows were aligned by BLASTX to the NCBI nr database and by BLASTN to the NCBI nt database. The maximum e-value allowed for the matches was 10−5. The regions that yielded significant matches to fungi were extracted and aligned back to the NCBI nr database to determine whether they were uniquely shared by H. monotropa and fungi.

The analysis of the origin of the cox1 intron in H. monotropa was performed as follows. First, we aligned its sequence to the NCBI nt database by using BLASTN online with the default parameters, taking the best 100 matching sequences. To these sequences, we added the sequences of all cox1 introns from Ericales that were not among the best 100 matches. Then, we aligned these sequences by using the MAFFT server at https://mafft.cbrc.jp/alignment/server/ (Katoh, Rozewicki & Yamada, 2019) with the default parameters. The unrooted phylogenetic tree of the cox1 introns was built by using RAxML 8.2.12 (Stamatakis, 2014) with 20 starting maximum parsimony trees and the GTR+Gamma model. The required number of bootstrap pseudoreplicates was automatically determined by the extended majority rule consensus tree criterion (the “autoMRE” option).

Phylogenetic analysis

Genes common to the mitochondrial genomes of 25 seed plants (atp1, atp4, atp8, atp9, ccmC, cob, cox1, cox2, cox3, matR, nad1, nad2, nad3, nad4, nad4L, nad5, nad6, nad7, nad8, nad9) were used for the phylogenetic analysis. Their sequences were concatenated and aligned with MAFFT (Katoh, Rozewicki & Yamada, 2019). The phylogenetic analysis was performed using RaXML (raxmlGUI v.1.3.1) with nucleotide sequences under the GTR+Gamma substitution model with 1,000 bootstrap replicates. To infer possible horizontal transfers, all protein-coding genes found in the H. monotropa mitogenome were aligned and analyzed in the same way as the concatenated gene set.

Results

Genome structure and gene content

Genome assembly resulted in two sequences. The GC content of both sequences is ~45%. One of the sequences is circular with a length of 106 Kb (Fig. 1). It does not exhibit any long repeats; the mapping of mate-pair reads showed an absence of pairs with abnormal insert sizes, suggesting that this sequence is not subject to recombination. The second, longer, fragment (Fig. 1) is ~704 Kb long. It was assembled into a linear sequence and exhibits a more complex structure. In particular, it contains several long repeats. As shown in Fig. 2, three pairs of long direct repeats were observed: the beginning of the chromosome and the 80–90 Kb region, the end of the chromosome and the 455–470 Kb region, the 415–420 Kb region and the 495–500 Kb region. Long inverted repeats were also found between the 260–262 kb region and the 657–659 kb region. A peculiar feature is a gradual decrease in the coverage on both of its ends (Fig. S1). This suggests that copies of the mitogenome with different lengths of this repeat coexist in plant cells. Repeats at the ends of the genome are observed in the linear mitochondrial genomes of several fungi, animals and protists (e.g., the works of Janouškovec et al. (2013) and Kayal et al. (2012)) and play an important role in their replication (Nosek et al., 1998). A similar mechanism could mediate the replication of the linear mitochondrial chromosome in H. monotropa. There are some mate-pair links between the two chromosomes, which suggests that they may recombine. However, the coverage of the smaller chromosome by mate-pair inserts (Fig. S1) presents no sharp drops; therefore, it is unlikely that recombination is frequent.

Figure 1 Maps of the mitochondrial chromosomes of Hypopitys monotropa.

Trans-spliced introns are indicated by three colored lines—red in nad1, green in nad2 and blue in nad5.

Figure 2 Repeats and mate-pair links in the mitochondrial chromosomes of Hypopitys monotropa.

(A) Repeats within and between the chromosomes. Direct repeats are connected by blue lines, inverted repeats are connected by orange ones. (B) “Improper” (with distance that exceeds expected length of the mate pair library or unexpected orientation) mate-pair links that indicate possible chromosome rearrangements. Read pairs with reads oriented in different directions (→ ← or ← →) are colored black and read pairs with reads oriented in the same direction (→ → or ← ←) are colored green. Only one of the two mate-pair libraries, that with the longer insert sizes (8,279 bp on average, standard deviation 2,583 bp), was used to build this diagram. A pair is considered improper if its reads are mapped not in the orientation → ←, or are mapped on different chromosomes, or are mapped in the orientation → ← but are separated by more than 20,000 bp. The green “torus” of closely situated reads in the orientations → → and ← ← comes from mate-pair reads improperly trimmed by NextClip, the tool for mate-pair reads’ trimming that we used.

To check for the presence of sequences transferred from fungi, we performed a BLAST search against all fungal sequences available from the NCBI. Although a BLASTX search identified several regions exhibiting similarity to hypothetical proteins from the Rhizophagus irregularis (Glomeromycota) genome, the same regions show high similarity to other plants as well (Table S1). This indicates that they are not the result of recent HGT mediated by mycoheterotrophy but, rather, are a result of either the ancient integration of fungal sequences into the genome of the common ancestor of Hypopitys and other flowering plants or the parallel integration of mobile genetic elements (e.g., mitoviruses) in the fungal and plant genomes. At least one of these regions exhibits high similarity to an RNA-dependent RNA polymerase gene from a known plant mitovirus, supporting the latter explanation.

Regarding gene content, the circular chromosome contains only two full-length protein-coding genes—ccmFc and cox1—and three exons of the nad5 gene, while two other exons are located in a linear fragment. Summary data for the annotation are presented in Table 1. All genes are supported by RNA-seq reads (Fig. S2; Table S2), and most regions with high coverage of RNA-seq reads correspond to annotated genes and their flanking regions (Fig. S3).

Table 1 Summary data on the structure and annotation of H. monotropa mitogenome.

Chromosome	Accession number (NCBI)	Length	Protein-coding genes	rRNA-coding genes	tRNA-coding genes	Pseudogenes	Genes with introns	Genes with cis-splicing	Genes with trans-splicing	
Larger (linear)	MK990822	704,088	35.5	3	17–18	2	8.5*	6	2.5	
Smaller (circular)	MK990823	106,028	2.5	0	1	0	2.5*	2	0.5	
Total			38	3	18–19	1	11	8	3	
Note:

* Fractional number of genes indicates that one gene has exons in both circular and linear fragments.

In general, the gene content of the H. monotropa mitogenome is typical of flowering plants. Surprisingly, it is even larger than that of V. macrocarpon, a close photosynthetic relative: atp6 and rps14 are pseudogenized and sdh3 and rps3 are absent in V. macrocarpon. There is only one pseudogene in H. monotropa, rps14. Pseudogenization or loss of rps14 is very common in flowering plants ((Figueroa et al., 1999)). In addition to the standard gene set, we found a new ORF (ORF671) that hypothetically encodes a 671-aa protein. It shows low similarity to a gene that encodes a hypothetical protein in the mitochondrial genomes of many flowering plants (the closest match is the hypothetical mitochondrial protein Salmi_Mp020 from Salvia miltiorrhiza). There are 4 genes encoding ribosomal proteins of the large subunit and 8 genes of the small subunit. Regarding the RNA component of the ribosome, all three ribosomal RNAs (26S, 18S, 5S) typical of plant mitochondrial genomes are present in H. monotropa. The set of tRNAs consists of 17 tRNAs that are typical of the mitogenomes of autotrophic plant species. There are no tRNAs of plastome origin except for tRNA-Gly-GCC. A selenocysteine tRNA (tRNA-Sec) gene and a sequence required for selenocysteine insertion during translation (SECIS element) have been reported in the mitogenome of V. macrocarpon (Fajardo et al., 2014). However, SecMarker, a novel specialized tool for searching for tRNA-Sec (Santesmasses, Mariotti & Guigó, 2017) that presents higher sensitivity and specificity than the tools used by Fajardo and coworkers, did not confirm the presence of a tRNA-Sec gene in the V. macrocarpon mitogenome. Thus, we infer that the predictions of a tRNA-Sec gene and a SECIS element in the V. macrocarpon mitogenome were false positives. In H. monotropa, we also did not find tRNA-Sec genes or SECIS elements. There is a region with moderate (78.8%) similarity to the presumed tRNA-Sec of V. macrocarpon. It is located upstream of ccmC as well as one of the two presumed tRNA-Sec sequences of V. macrocarpon. tRNA genes are known to exhibit highly conserved sequences; in particular, the similarity of other V. macrocarpon and H. monotropa tRNA genes is 98–100%. This suggests that the presumed tRNA-Sec of V. macrocarpon is not a functional gene but a pseudogene of a tRNA-coding gene.

Despite being highly conserved in coding sequences, mitochondrial genes sometimes differ in their intron content. For example, cox2 may consist of three exons (D. carota, V. vinifera), two exons (C. paramensis, A. thaliana), or a single exon (V. macrocarpon). In many angiosperm lineages, the cox1 gene contains a group I intron, presumably acquired via multiple horizontal transfer events (Cho et al., 1998; Sanchez-Puerta et al., 2008, 2011), although there is an opposing hypothesis indicating a single HGT event and multiple losses (Cusimano, Zhang & Renner, 2008). The cox1 intron is highly overrepresented in the parasitic plants that have been examined to date (Barkman et al., 2007), although the hypothesis that parasitism may serve as a mediator of horizontal intron transfer is not supported by phylogenetic analysis (Fan et al., 2016). V. macrocarpon lacks an intron in cox1, while in H. monotropa, an intron is present. Phylogenetic analysis indicates that the intron found in H. monotropa has the same origin as that in Pyrola secunda (Fig. S4) and was thus presumably vertically inherited from the common ancestor of Pyrola and Hypopitys. In other genes, H. monotropa exhibits the same intron content as V. macrocarpon. Three genes—nad1, nad2, and nad5—have trans-spliced transcripts. Notably, the nad5 exons are located in different chromosomes—exons 1, 2 and 3 are located in the linear chromosome and exons 4 and 5 in the circular one—suggesting a case of interchromosome trans-splicing, which has been reported in other plants that have multichromosomal mitogenomes (Lloyd Evans et al., 2019; Sloan et al., 2012).

Detailed data on the genes and their intron-exon structure and expression are presented in Table S2.

Sequences transferred to the mitogenome

In the mitochondrial genome, we found fragments with high similarity to plastid genes, including both genes that are present in the H. monotropa plastome (matK, rps2, rps4) and genes that have been lost (rpoB, C1, C2, psbC, ndhJ, B, ycf2) (Table S3). These fragments were derived from horizontal transfers from plastids (MIPTs; presumably intracellular). Such events are frequent in plant mitochondria (for example, see the works of Alverson et al. (2011) and Grewe et al. (2014)). In rarer cases, interspecies horizontal transfer from the mitochondrial genomes of other plants harboring inserts from the plastid genome may be a source of MIPTs, which is referred to as foreign MIPTs (Gandini & Sanchez-Puerta, 2017). MIPTs in nonphotosynthetic plants are of particular interest. If mitochondrial copies of plastid genes that were lost or pseudogenized in the plastome retain intact ORFs, the reverse switch from heterotrophy to autotrophy is potentially possible. The mitochondrial genome in this case acts as a “refugium” of plastid genes. The Castilleja paramensis mitogenome contains 55 full-length or nearly full-length plastid genes, and only approximately half of them are obvious pseudogenes (Fan et al., 2016). Currently, the only example of functional mitochondrial genes transferred from plastids is provided by the tRNA genes of plastid origin recruited for protein synthesis in mitochondria (Joyce & Gray, 1989). In H. monotropa, none of the sequences that originated from plastid protein-coding genes present intact ORFs, retaining less than 50% of the initial length and/or carrying multiple frameshift mutations in most cases (Table S3).

Mitochondrial gene expression and RNA editing

To gain insights into mitochondrial gene expression and RNA editing and refine the sequence annotation, we sequenced and assembled the transcriptome of H. monotropa (Logacheva et al., 2016; Schelkunov, Penin & Logacheva, 2018). We observed the expression of all annotated protein-coding genes (hereafter, the expression level is defined as the number of mapped read pairs divided by gene length). A minimal expression level (0.1) is observed for the hypothetical protein ORF671. Cytochrome c maturation factors are also expressed at low levels (3 of 4 genes have expression <1). The genes of Complex I exhibit intermediate expression (1–6). The highest expression is observed for sdh4, cox3, atp1 and atp9 (Table S2).

RNA editing is a phenomenon that is widely observed in mitochondrial transcripts. The number and position of editing sites differ greatly from species to species and from gene to gene. In H. monotropa, we identified 545 RNA editing sites present in at least one RNA-seq sample (Table S4) under an editing-level threshold of 10% and a coverage threshold of 20×. Most of these editing events are observed in all samples (Fig. 3), and most of them are C–U events. However, we found at least one A–G editing event, which was present in all samples at the level of ~20% (see “Discussion”). The transcripts of 33 genes out of 38 are edited; the maximal RNA editing density (measured as the number of sites per 100 bp) was observed for ccmB. The genes of Complex I exhibit different RNA editing densities, ranging from 0.2 for nad5 to 5.6 for nad4L. In nad4L, RNA editing restores the start codon; presumably, the same is true for V. macrocarpon, in which nad4L has been (mis) annotated as a pseudogene. The RNA editing density in the genes of Complexes II and III is low, not exceeding 1.1. Finally, in atp1, rpl16, rps12, 13 and ORF671, we did not find any editing events. Several genes with high expression present low levels of RNA editing (atp1, rps12) and vise versa (cytochrome c maturation factors), but there is generally no such tendency. The median dEN/dES value across all genes is 1.23. There are no edited stop codons (which is expected because most of the editing events are C–U events and stop codons lack C; however, the observation of non-C–U editing potentially enables this), but there are several stop codons that are introduced by editing (Table S5). To compare RNA editing in H. monotropa with that in other species, we used data from A. thaliana and O. sativa, the model plants in which RNA editing has been thoroughly characterized (Giege & Brennicke, 1999; Notsu et al., 2002). Overall, there are 842 positions that are edited in at least one species among the protein-coding genes common to these three species (Table S6). Only 151 (17.9%) of these positions are edited in all three species. Notably, 37.9% of the positions are that are edited in at least one species contain a T in the species that do not exhibit editing in this position, while only 6.1% contain an A or G. This is congruent with the results of (Edera, Gandini & Sanchez-Puerta, 2018), who found that the main pattern of the loss of RNA editing is the replacement of editing sites with thymines. A total of 337 (38.1%) positions contain unedited cytosines.

Figure 3 Venn diagram representing the occurrence of RNA editing events in different RNA-seq samples.

Phylogenetic analysis

As mentioned above, plant mitochondrial genomes are prone to HGT. It is often detected from the incongruence of phylogenetic trees based on different genome regions (Bergthorsson et al., 2004; Cusimano & Renner, 2019). We performed phylogenetic analysis of individual mitochondrial genes and found that the topologies were similar with regard to the placement of H. monotropa—it is always placed together with V. macrocarpon (excluding the cases of unresolved nodes) (see Fig. S5). This evidence indicates that no genes were acquired via HGT. The combined tree of all mitochondrial genes shared across 25 seed plant species shows a topology similar to that based on nuclear and plastid genes, with monocots representing a monophyletic group and eudicots divided into two large groups—asterids and rosids. H. monotropa is sister to V. macrocarpon, and both species are within the asterids, as expected (Fig. 4). Notably, H. monotropa genes do not exhibit any increase in substitution rates. The same is true for another MHT plant, Petrosavia stellaris (data from (Logacheva et al., 2014)). Parasitic plants were previously reported to exhibit elevated substitution rates in all three genomes (Bromham, Cowman & Lanfear, 2013); however, a recent study involving broader sampling of parasitic plants shows that this is not a universal phenomenon (Zervas, Petersen & Seberg, 2019).

Figure 4 Phylogenetic tree based on the maximum likelihood analysis of nucleotide sequences of the 20 genes set.

Values above nodes indicate bootstrap support. Branch lengths are proportional to the number of substitutions.

Discussion

Mitogenome structure

All available evidence suggests that the H. monotropa mitogenome contains a linear fragment. While linear plasmids are found in the mitochondrial genomes of several plants (e.g., in the work of Handa, Itani & Sato (2002)), the linear fragment of the H. monotropa mitogenome lacks characteristic features of these plasmids such as a terminal inverted repeat, small size (not over 11 Kb) and genes of RNA and DNA polymerases (reviewed in Handa (2008)). A linear structure is typical for the mitochondrial genomes of fungi and protists (Janouškovec et al., 2013; Nosek et al., 1995). Similar to other mycoheteroprophic plants, H. monotropa lives in intimate symbiosis with fungi (Min et al., 2012). One might hypothesize that the linear fragment could be the result of either contamination or HGT from fungi. However, these explanations are unlikely to be correct for the following reasons: (1) DNA was isolated from inflorescences, while mycorrhizae exist only in roots; (2) all potential fungal hosts of H. monotropa with a known mitogenome possess a circular chromosome and no linear plasmids; (3) there are no fragments with similarity to known fungal genomes; and (4) all genes annotated in the linear fragment are typical plant mitochondrial genes. The observation that a single circular molecule (“master circle”) is an oversimplified representation of the plant mitochondrial genome and that these genomes instead exist in vivo as a mixture of circular, linear and branched forms is not novel (see, for example, the report of Sloan (2013)). However, a circular structure can usually be observed at the level of sequence assembly due to the presence of multiple repeats. This is not the case for H. monotropa, in which internal repeats are also present, but their location and the distribution of mate-pair links do not allow us to reconstitute the master circle (see Fig. 2). This suggests that the diversity of the organization of plant mitochondrial genomes could be even greater than that reported recently (Kozik et al., 2019). An alternative explanation for the observed linear structure is that this chromosome contains a very long (longer than the size of mate-pair inserts) highly GC-rich region that cannot be sequenced using Illumina technology. Third-generation sequencing technologies, particularly the Pacific Bioscience SMRT platform, are able to handle such regions (Loomis et al., 2013). However, we consider this explanation to be unlikely for two reasons: first, plant mitochondrial genomes are universally biased towards AT nucleotides, and second, we would have observed GC-rich regions at the ends of the linear fragment in this case, which were not found.

A high level of convergence is observed in the gene set of plastid genomes of non-photosynthetic plants, regardless of their parasitic or MHT status. They are characterized by a certain degree of reduction, which usually correlates with the timing of the transition to heterotrophy (for example, see the report of Samigullin et al. (2016)) and follows the general gene loss model (Barrett et al., 2014). In contrast, the mitogenomes of heterotrophic plants are very diverse in terms of their structure, size and gene content. In H. monotropa, the total size of the mitogenome is 810 Kb, which is almost twice as large as that of V. macrocarpon. However, this expansion is unlikely to be associated with heterotrophy. Large mitogenomes are known from heterotrophic plants, particularly from the MHT orchid Gastrodia elata (~1.3 Mbp) (Yuan et al., 2018) and parasitic Cynomorium (1 Mbp) (Bellot et al., 2016) and Lophophytum mirabile (Balanophoraceae) (~820 Kb) (in the latter case, the size is shaped by fragments horizontally transferred from the host—see below). The other extreme is represented by the highly miniaturized mitogenomes of V. scurruloideum (Skippington et al., 2015).

Horizontal gene transfer

HGT is a very common phenomenon in parasitic plants (Yang et al., 2016). HGT from host plants into the mitochondrial genome has been shown in Rafflesiaceae (Xi et al., 2013), Orobanchaceae (Yang et al., 2016), Cynomorium (Cynomoriaceae) (Bellot et al., 2016), and Lophophytum mirabile (Balanophoraceae). In the last case, horizontally transferred homologs have replaced almost all native mitochondrial genes (Sanchez-Puerta et al., 2017, 2019). In contrast, there are no traces of HGT in the H. monotropa mitogenome. The nuclear and mitochondrial genomes of the MHT orchid Gastrodia elata were recently characterized (Yuan et al., 2018), as was the mitochondrial genome of the MHT dicot Epirixanthes elongata (Petersen et al., 2019), and no HGT was observed in these cases. Although MHT plants are usually regarded alongside the parasitic plants that feed on other plants, the interactions between plants and their hosts are very different in these two cases. Parasitic plants develop specialized structures that integrate into the vascular system of a host plant and channel the flow of nutrients from the host to themselves. Such connections are similar in many aspects to graft junctions and can be the route of transport not only for nutrients but also for high-molecular-weight compounds, including proteins and nucleic acids. The bidirectional transfer of nucleic acids through haustoria in the parasitic plant Cuscuta pentagona has been shown (Kim et al., 2014). The transfer of RNA from the host is hypothesized to mediate HGT into the parasite genome. In contrast, such transfer is not known to occur in mycorrhizal symbioses. This emphasizes that despite similar heterotrophic strategies, plant parasites and MHT plants are fundamentally different in terms of the interaction with their hosts and, potentially, many other features that stem from this interaction. This calls for the increased sampling of MHT plants in genomic projects and for the development of new model systems representing MHT plants. Many of these species are rare endangered plants with very small distribution ranges; in contrast, H. monotropa is widespread and is thus presents potential as a model MHT plant.

RNA editing

RNA editing is an important characteristic of mitochondrial gene expression. It varies greatly in plants from complete absence in the bryophyte Marchantia polymorpha to the occurrence of several hundred or even thousands of editing events (for a review, see Ichinose & Sugita (2016)). RNA editing in nonphotosynthetic plants is of special interest because many proteins are involved in RNA editing in both plastids and mitochondria (for example, see Bentolila et al. (2012)). Thus, the reduction of the plastid genome and the coordinated loss of nuclear genes involved in editing in plastids can also influence mitochondrial RNA editing. Currently, data on RNA editing in mitochondria for nonphotosynthetic plants are scarce. C-to-U RNA editing was found in seven genes (atp1, atp4, atp6, cox2, nad1, rps4 and rps12) in R. cantleyi (Xi et al., 2013); and Barkman and coauthors demonstrated editing in the putatively horizontally transferred atp1 gene in a wide range of plant species (Barkman et al., 2007). In V. scurruloideum, C-to-U editing was predicted computationally for the nine protein-coding genes (Skippington et al., 2015). In H. monotropa, we observed editing in the majority of genes; almost all editing events were of the C-to-U type, which is typical of plant mitochondria. The vast majority of the editing events were found in all samples (Fig. 3; Table S4). We observed a single non-C-U editing site, as A–G editing was observed in the rps19 gene transcript. This editing was represented in all samples (inflorescence, ovules and anthers) at the level of 17–23% of reads (Table S4). Considering that there was no coverage peak at this position, it is unlikely that this was a result of read mismapping. We hypothesize that the A–G conversion that we observed actually represent not A-to-G editing but adenosine-to-inosine editing (inosine reads as G in sequencing data). This type of editing is typical of animals and fungi (Cattenoz et al., 2013; Liu et al., 2016) but has not previously been found in plants. In many cases, RNA editing plays a clear functional role (e.g., restoration of typical start codons in the plastid genes rpl2 and psbL (Kudla et al., 1992)). We found one such case in H. monotropa (nad4L). A more detailed examination of RNA editing in MHT plants, including its dynamics in different organs and developmental stages, is required to highlight potentially functional events.

Conclusions

Nonphotosynthetic plants represent ~1% of plant diversity and provide excellent models for the study of convergent evolution. Until recently, the genomic research in non-photosynthetic plants was focused on plant parasites; a common assumption is that mycoheterotrophs—plants that parasitize fungi—exhibit basically the same patterns of genome evolution as plant parasites. To test this hypothesis and to expand our knowledge of MHT plants, we characterized the mitochondrial genome of H. monotropa. Additionally, using RNA-seq, we performed a genome-wide analysis of gene expression and RNA editing. We showed that the mitogenome structure of H. monotropa is highly unusual: it includes a small circular fragment and a large linear fragment with multiple repeats at its ends that presumably function as telomeres. Further studies that include the characterization of mitogenomes of other Ericaceae and the in vivo analysis of H. monotropa mitochondria are required to investigate the details of the evolution, replication and functioning of such unusual mitogenomes. The gene set is similar to that of autotrophic plants. All protein-coding genes are expressed, and in most of these genes (33 out of 38), we observed editing of the transcripts. The intergenic regions of the mitogenome carry multiple sequences of plastid origin, including sequences of photosynthesis-related genes that are absent in the H. monotropa plastome. We did not find any traces of HGT from fungal hosts in the H. monotropa mitogenome or an increase in nucleotide substitution rates. These new data highlight the contrast between MHT and parasitic plants and emphasize the need for new model species representing MHT plants.

Supplemental Information

Supplemental Information 1 Results of the BLAST search of regions that have significant similarity to fungal genes.

Click here for additional data file.

Supplemental Information 2 Characteristics of mitochondrial genes of H. monotropa: exon-ontron structure, position, expression.

Click here for additional data file.

Supplemental Information 3 Plastid genes in mitogenome of H. monotropa.

Click here for additional data file.

Supplemental Information 4 RNA editing of H. monotropa mitochondrial genes.

Click here for additional data file.

Supplemental Information 5 Genes with stop codons that are introduced by editing.

Click here for additional data file.

Supplemental Information 6 RNA editing of H. monotropa mitochondrial genes compared with Arabidopsis thaliana and Oryza sativa.

Click here for additional data file.

Supplemental Information 7 Read mapping characteristics along the mitochondrial chromosomes of Hypopitys monotropa.

(A) Average insert size between mate-pair reads spanning over different genomic positions. The nearly uniform insert size distribution suggests that there are no misassemblies involving large deletions or insertions. The fluctuations at the ends of the larger chromosome result from small numbers of reads mapping to the very ends of the chromosome, which is linear. (B) Number of mate pair fragments covering each of the chromosomes’ positions. The absence of positions with zero coverage suggests that there are no misassemblies involving genome fragments’ rearrangements. The drops on the ends of the larger chromosome follow from its linearity. (C) Coverage of the chromosomes by reads of all three sequencing libraries: the paired-end library and both mate-pair libraries. The coverage, though fluctuating, never reaches zero, thus suggesting the absence of misassemblies. Coverage at the ends of the smaller (circular) chromosome abruptly drops approximately sixfold, due to difficulty of mapping reads part of which map to the end and part to the beginning of the contig. In the larger (linear) chromosome, the gradual drops of coverage near the edges and near the positions 90,000 bp and 450,000 bp are due to the varying repeat copy number (see discussion in the main text).

Click here for additional data file.

Supplemental Information 8 Coverage of H. monotropa mitochondrial CDS by RNA-seq reads.

Click here for additional data file.

Supplemental Information 9 Coverage of H. monotropa mitochondrial chromosomes by RNA-seq reads.

Click here for additional data file.

Supplemental Information 10 Phylogenetic tree of cox1 intron. Branches with bootstrap support below 70 are collapsed.

Click here for additional data file.

Supplemental Information 11 Phylogenetic trees inferred from ML analysis of single mitochondrial genes. Branches with bootstrap support below 50 are collapsed.

Click here for additional data file.

The authors thank Alexey Kondrashov (University of Michigan, Ann Arbor) for providing plant material and Artem Kasianov for assistance with the data analysis.

Additional Information and Declarations

Competing Interests

Author Contributions

Field Study Permissions

DNA Deposition

Data Availability

The authors declare that they have no competing interests.

Viktoria Yu Shtratnikova analyzed the data, prepared figures and/or tables, authored or reviewed drafts of the paper, and approved the final draft.

Mikhail I Schelkunov analyzed the data, prepared figures and/or tables, authored or reviewed drafts of the paper, and approved the final draft.

Aleksey A. Penin conceived and designed the experiments, performed the experiments, prepared figures and/or tables, and approved the final draft.

Maria D. Logacheva conceived and designed the experiments, performed the experiments, analyzed the data, prepared figures and/or tables, authored or reviewed drafts of the paper, and approved the final draft.

The following information was supplied relating to field study approvals (i.e., approving body and any reference numbers):

Hypopitys monotropa is not an endangered or threatened plant species and thus according to Russian laws and local rules of our university no specific permissions were required for collection. The land is public so no permission is applicable.

The following information was supplied regarding the deposition of DNA sequences:

The sequences are available at GenBank: MK990822 and MK990823 (BioProject: PRJNA522958 and PRJNA573526).

The following information was supplied regarding data availability:

Raw data is available at NCBI, Bioproject PRJNA522958.

The code of the scripts written by the authors of this article and used for the analyses is deposited on Figshare. The link is DOI 10.6084/m9.figshare.8247023.

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
