# Peer review of "Mitochondrial genome of the nonphotosynthetic mycoheterotrophic plant Hypopitys monotropa, its structure, gene expression and RNA editing"

_PeerJ, doi:10.7717/peerj.9309_

## Round 0.1 · original submission · Major Revisions

· Academic Editor

Major Revisions

The reviewers generally find much to like about the manuscript and I recommend you follow their suggestions with one exception. Reviewer 1 notes that "Supplementary Figure S1C legend: the authors stat:e “Coverage of the chromosomes by reads of all three sequencing libraries: the paired-end library and both mate-pair libraries” What is the difference between a paired-end library and a mate-pair library? I believe there are no differences and the three libraries should be distinguished by the insert size or any other variable feature." There are differences between mate pair and paired-end libraries and I assume that you used those words appropriately. Please confirm this in your response to that comment.

Reviewer 1 ·

Basic reporting

Raw data: The SRA accession number of the data is already accessible because it was mentioned in a previous publication.
The accession numbers of the mitochondrial genome of H. monotropa are included in Table 1 but are not yet accessible. Given the number of errors in Genbank data bases on angiosperm mitochondrial genomes because reviewers are normally not able to access the deposited data in advance, I suggest that the data be accessible to reviewers at this stage.

Even though the English language was generally clear and unambiguous, some sentences should be improved to clearly understand your text.
Some examples where the phrasing could be improved include lines 76-78, 128-131, 289-291.
Some examples where I identified grammatical mistakes include lines 50-51, 54-55, 293-294, 310-311.

The introduction includes appropriate context and the literature is up to date and relevant. A few comments regarding the Introduction:

Lines 40-41: in Bjorkman 1960 this species was named as Monotropa hypopitys, if the name was changed or if it is a case of synonymy it should be noted.

Lines 67-69: the authors mentioned the mitogenomes of Rafflesiaceae and Cynomoriaceae as examples of massive HGT. However, the mtDNA of Cynomorium does not show massive HGT, while the holoparasite Lophophytum mirabile does and was not mentioned (Sanchez-Puerta et al 2017 New Phytol, 2019 Mol. Phylogenet. Evol). Also, citations of these studies are missing.

Line 74: ccmC, ccmFc and ccmFn are annotated as functional (but very divergent) genes in Viscum species

Lines 86-87 and lines 224-225: the gene nad4L is not a pseudogene in Vaccinium because the ORF is intact and it only needs RNA editing in the first codon to go from ACG to ATG. Otherwise, it is well-conserved and of the expected length.

Experimental design

The research questions are well defined, relevant and meaningful. It is stated how the research fills an identified knowledge gap. The methods are described with sufficient detail in general. A few comments on the experimental design:

Lines 92-94: Relevant DNA and RNA library details should be included here along with the citation of the fully described protocol. For example, important details include the library insert size, read length, paired-end reads or not, size of sequence data.

Line 95: Given that plant mitochondria normally contains about 50% of its sequence with no homology to any sequence in the NCBI databases, what did you do to make sure that mitochondrial contigs with no hits to Vaccinium mtDNA were included in the final mitochondrial assembly?

Lines 101-102: I recommend doing BLASTN searches against all mitochondrial genome plants, not only Vaccinium macrocarpon for the first screening of mitochondrial contigs. Otherwise, you might be missing some mitochondrial contigs.
Lines 109-111: What are the coverage’s intervals that you considered as nuclear, mitochondrial or chloroplast?
What was the coverage of nuclear contigs?

Lines 120-138: The term chromosome should be replaced by contigs because there is not enough information to support the presence of two autonomous chromosomes. I suggest using contigs or molecules to refer to the circular and linear assemblies of the MT.
Line 129-130: The explanation for the decrease in coverage at the ends of the linear contig is not clear. In fact, I believe that it is normal to see a decrease at the ends due to the alignment of reads, as stated in the legend of Figure S1B “The drops on the ends of the larger chromosome follow from its linearity.”
Lines 147-148: if the read depth is used to suspect unnoticed protein-coding genes, then the alignment of the RNAseq reads should be more stringent (not relaxed) because lots of noisy reads could result in an artefactual high coverage.

Lines 163-164: dN/dS refers to the number of nonsynonymous and synonymous changes per nonsynonymous and synonymous sites, respectively. Therefore, the calculation done here is not really similar to dN/dS.

Lines 168-169 the mitochondrial plastid DNA are commonly known as MTPTs, I don’t see the need to introduce a new term (MIPT) for the same feature.
Lines 169-171: why only the genes of Cammellia and not the intergenic regions, that is, the whole cpDNA was aligned to the H. monotropa mtDNA? It is common to see MTPTs derived from plastid intergenic regions.

169-170: I suppose you are talking about Camelia plastid genes. Please clarify in the text.

Lines 181-183 and 262-263: based on the alignments produced here, it would be simple and interesting to run phylogenetic analyses and test whether any of these MTPTs were foreign. It is very common in plant MT to find foreign MTPTs (see Gandini et al 2015 Sci. Rep.). In particular, it would be relevant in this case that a few plastid genes are not present in the plastid genome of H. monotropa. Therefore, if these genes are native, it would indicate an ancestral plastid-to-mitochondrion intracellular gene transfer. If foreign, there could be more recent cases of horizontal gene transfer.

Lines 184-189: It would be better to run BLAST searches against nr instead of restricting the search to Russula brevipes. BTW, is Russula brevipes a fungus normally associated to H. monotropa? Do they have a specific relation or is H. monotropa generalist? Please clarify the rationale behind your limited search criteria.
In addition, searching against Russula brevipes gives unsufficient information for the goal state here.

Lines 191-194 and 303-305: were editing sites removed from the alignments before running the
phylogenetic analyses of individual genes? To validate the statement that all the topologies were identical, please include the individual gene phylogenies as supplementary.
How many RAxML bootstrap replicates were done?

Validity of the findings

The results are meaningful, although a little disorganized and unclear at times. The conclusions are generally limited to supporting results. See the comments on the Results and Discussion sections below:

Lines 200-201 belong to the Methods section.

Supplementary Figure S1C legend: the authors stat:e “Coverage of the chromosomes by reads of all three sequencing libraries: the paired-end library and both mate-pair libraries” What is the difference between a paired-end library and a mate-pair library? I believe there are no differences and the three libraries should be distinguished by the insert size or any other variable feature.

Lines 236-237. Given that H. monotropa has the cox1 intron and Vaccinium does not, it indicates that it has been acquired by HGT. It would be worth including a tree of the cox1 intron to find out its phylogenetic affiliation.

Lines 239-240: nad5 has 5 exons. Please search for exon 3 which is only 22bp long.

Lines 240-245: interchromosome trans-splicing has been previously reported in multichromosomal mitochondrial genomes such as Silene and Lophophytum. In addition, I don’t consider it a rare event, as much as the rarity of the presence of multichromosomal mtDNAs.

Lines 270-272: Plastid genes that are missing in the plastid genome could be lost from the plant or could have been transferred to the nuclear genome. I believe this possibility or the opposite ( if the absence of these missing genes in the nuclear genome is confirmed) should be included in the discussion.

Lines 281-287: How was the differential expression of the mitochondrial genes evaluated? If the relaxed alignment method described in Methods was used, it could be artefactual. To avoid misleading conclusions, the alignments of RNAseq data on each gene should be visualized to detect the mismapping of reads. In addition, a figure showing the read depth of the RNAseq data over the mitochondrial genome should be included.

Line 291 “most of them are C-U”. Given that only C-U editing is known in plant mitochondria, any case of a different editing type should be clearly stated, verified and reported. If not, the mismatch should be explained by alternative hypothesis.

Lines 297-298: stop codons cannot be edited because none of the 3 stop codons lack any cytosine.

Suppl Table 1: it should include each editing site, including the change, if different from C-U, whether it is synonymous or nonsynonymous and the editing extent of each site. The bulk or average numbers provided in this table are not very informative.
Did you identify edition of RNAs and the sequences that surrounding protein-coding regions? Are intergenic regions edited?


329-337: the fact that the authors assembled a linear molecule does not mean that it exists in vivo. It could be the result of missing sequences in the assembly that would allow the assembly of one or more circular molecules (see comment above on the assembly methods). Please comment on this or provide a possible explanation for the replication and maintenance of this linear molecule.
The title mentions the unusual structure; however, this should be revised because most angiosperm mtDNAs are lineal in Genbank and published as such due to the complexity in the assembly process and not necessarily due to a real replicating-structure in vivo.

360-365: The fact that this mycoheterotrophic plant and also Gastrodia do not show any fungal sequences despite their intimate relationship could be explained by the “mitochondrial compatibility fusion model” of plant HGT described in Rice et al 2013 Science”. Fungal mitochondria cannot fuse with plant mitochondria and this is the hypothetical mean for plant to plant HGT.

Line 368: what is the meaning of MHT plants? It should be defined the first time this term is used.

373-386: the discussion of the editing sites is incomplete. For example, much has been discussed on the editing of synonymous sites and it correlation with low editing extent or efficiency. Also, the statistical dEN/dES= 1.23 is not well explained and thus, it is difficult to interpret (see comments above).
Also, the phrase in the title “extensive RNA editing” is somewhat misleading because knowing that plant mitochondria hace extensive RNA editing, it seems that this plant shows particularly extensive RNA editing, which is not the case.

395-396: the presence of repeats at the end of the linear molecule is also compatible with assembly problems or artifacts. Assemblers have trouble when repeated sequences are present and tend to produce contigs and stop at repeats.

Figures and tables are relevant, high quality, and appropriate. However, there are a few suggestions for improvement:
Table 1 indicates the number of genes in each mt chromosomes (should be replaced by contigs or molecules) but it is really the “number of protein-coding genes” as it is not considering tRNAs or rRNAs. Please modify. 5SrRNA is missing in the Table.

Figure 1: There are mistakes in gene color assignment (e.g. ccm genes are in violet instead of being in dark-green). I don’t see the rRNA 5S. Repeats and MTPTs should be included. Trans-spliced lines are not needed.

Figure 3: the phylogenetic tree is rooted in Liriodendron but it should have been rooted in Cycas because it is the only non-angiosperm present.

Additional comments

The results, that is, the report and analysis of the mtDNA of H. monotropa is original, interesting and worthy of publication. However, the assembly, the analyses and text should be severely improved. The data should be analyzed in more detailed, making better and more in-depth interpretations to reach solid conclusions and make statements that are clearly supported by the data and the bibliography. I hope that the detailed comments I provide are helpful to the authors.

Reviewer 2 ·

Basic reporting

This study focuses on the assembly of mitochondrial genome of H. monotropa and analyses regarding mitochondrial RNA editing and horizontal gene transfer.However, this article has major problems in their experimental design and the following analyses. If done correctly, this study would be a good resource for the field. However, this manuscript is not ready to be published before the following corrections have been made and raw data deposited.

Experimental design

Assembly:
a.The authors used the mitogenome of V.macrocarpon as the reference for selecting mito sequences from H. monotropa because the mito genome of the two species are ‘phylogenetically closest’. However, the authors did not provide any evidence supporting their ‘phylogenetically closest’ relationship. Although in their phylogenetic analysis later, they are in the same clade, but this may be simply due to the fact that V.macrocarpon was used as the reference for assembly. In addition, V.macrocarpon is a photosynthetic plant whereas H.monotropa is not. In sum, it seems reckless to use only one species as the reference genome for assembly. The authors should use more species as the reference.
b. It seems that the authors excluded the cp sequences for further analysis in the assembly of mito genome (L108). However, this is not reasonable due to ‘the presence of inserts from the cp genome into the mito genome. It has been reported in many species that cp and mito genome share similar sequences. Furthermore, it is unclear how the authors mapped reads to nuclear contigs (which organism was used as reference?). Thus, it is hard to evaluate if they did it correctly in removing nuclear sequences.

HGT:
a. The authors attempted to identify the sequence that transferred from the plastome to the mitogenome. However, they excluded the cp sequences for further analysis in their sequence assembly. Therefore, this analysis does not make any sense. Second, they used Camelia sinensis’s sequence as the reference genome in this step. In the results, they said they intended to look for ‘intracellular’ horizontal transfers. For intracellular transfers, the H.monotropa’s cp genome should be used as the reference. In addition, their claim that C.sinesis is closest to H.monotropa does not have a reference.
b. For examining possible horizontal gene transfers from fungi, the authors used sequences from R. brevipes. However, the assembled mito sequence was based on a blast to the V. macrocarpon genome. Given that V.macrocarpon is a photosynthetic plant, it is less likely to contain sequences from R.brevipes. Therefore, sequences from fungi would be eliminated in a prior step, thus they did not find any horizontal gene transfer from fungi is expected.

RNA editing:
a. RNA-seq of at least two bio reps needs to be performed for identification of RNA editing sites for statistical significance.
b. How the expression level (L284) is derived is a mystery.
c. ‘Most of them are C-U. Was there other types of editing? Please clarify.
d. “Level of editing’ should refer to the editing extents (up to 100%), not how many sites are edited in 100 bp.
e. Please provide a full list of all editing sites and corresponding editing extents.

Validity of the findings

No comment

Additional comments

No comment

Reviewer 3 ·

Basic reporting

Basic Reporting: Although this is an interesting plant to study, the introduction section would benefit from some reorganization, revisions and better review of background literature.

For instance, the concluding paragraph of the Introduction on lines 83-89 are spent introducing the Vaccinium mitogenome. This would be better placed before the section of the introduction that outlines the objectives of the project.

On lines47-49 the authors refer to most attention focused on plastid genomes of mycoheteroptrophs but only refer to two studies on Monotropa (2016 but still “in press” in the Lit. cited?) and a second study of their own. This is really a biased and insufficient treatment of a vast literature.

In lines 52-57 the authors need to improve the writing because in the first sentence they state that plant mitogenomes are highly variable in size and gene content, yet in line 57 they state that gene content is “rather stable”. Since this is a topic of most relevance to the rest of the paper, this paragraph probably needs to be improved.

On line 62 the authors refer to non-photosynthetic plants as being either parasitic and mycoheterotrophic. There are also photosynthetic parasites and mycoheterotrophs so I am not sure why these are excluded from the literature review here.

Experimental design

Experimental design: The methods used are largely clear and seem robust. It is really excellent that the authors used RNAseq to compare to the genomic sequences obtained.

It would be beneficial for the authors to justify why this particular mitogenome is important to study. There is already one mycoheterotroph studied. Is it a particularly good representative to estimate HGT and study RNA editing? If so, why?

Lines 120-138: It might have been useful to use PCR to attempt to experimentally verify contig assemblies and chromosome structure.

Lines 160-166: It is entirely unclear to me why the authors are studying RNA edit sites. I personally find RNA editing to be fascinating but no justification was given for why it is investigated in this study.

Line 184-189: It is not clear that a rigorous investigation of HGT can be achieved if sequences are only compared to Russula. Couldn’t there be other donors?

Validity of the findings

Validity of the findings: Largely, I think that the results are justified and nicely presented. However there are shortcomings.

Lines 220-223: The authors claim that “Unlike parasitic plants, which show a certain degree of reduction, from limited to high….the gene content in H. monotropa is not reduced.” It is not clear to me how the situation for H. monotropa is unlike parasitic plants since some parasitic plants show gene content that is not reduced as well.

Lines 225-228: The authors indicate that pseudogenes exist within the H. monotropa mitogenome but we are never told what the basis for that determination is. This should either be explained here or in the methods section.

Lines 229-261: This section needs help. First, the authors discuss group I introns, cis and trans-spliced introns in a non-organized way that is hard to follow and then the rest of the paragraph is focused on ribosomal RNA and tRNA sequences. Probably this would benefit by being separated into different paragraphs.

Lines 231-238: The authors probably need dig more carefully into the coxI intron literature, especially as it relates to horizontal gene transfer and also parasitic plants.
First, the Cusimano 2008 paper has some valuable contributions regarding some degree of vertical inheritance of the intron, especially in Araceae and other have shown a case of limited vertical inheritance in orchids but this paper is flawed in other ways as shown convincingly by Sanchez-Puerta et al., 2008.
Second, the authors should cite their source for “The cox1 intron is highly overrepresented in the parasitic plants…”. Presumably that is Barkman et al., 2010?
Third, the authors only compare Vaccinium and H. monotropa for coxI intron presence; yet, numerous Ericales species have the intron, perhaps of most relevance is Pyrola since it is also in Ericaceae.
Fourth, the authors claim there is no evidence for HGT in the H. monotropa mitogenome but never show a phylogenetic analysis for this coxI intron. A phylogenetic analysis is needed to show what other species the intron is closely related to. If Pyrola, perhaps it was vertically inherited. If not, it is likely present due to HGT and therefore the claim on line 356 needs to be revised.

Line 241-243: The authors report a new ORF but suggest that it has no significant similarity to other sequences. It would be better to show the evidence for this. Trying lower stringency would be important. It is also not clear if it has no similarity to ANY sequence in GenBank or if it just doesn’t match sequences in mitogenomes.

Lines 265-267: If these plastid sequences are due to IGT then they should be shown by phylogenetic analysis to be closely related to the same genes from the H. monotropa plastid genome. A phylogenetic analysis needs to be performed to show this.

Lines 284-287: It is not clear from the results or methods what the expression levels are based upon. FPKM?

Lines 288-299: The authors should compare their reported edit sites to those that are known from other plants. Usually these show conservation across species. Although atp1 is listed as having zero edit sites, it has been reported to show editing at a few sites.

Lines 301-314: The authors suggest that all single genes and combined genes give the same topology. There are no data shown to substantiate this claim. I would be fairly surprised if this were to be the case. RNA edit sites can have a huge impact on phylogenetic reconstruction as demonstrated by Bowe and dePamphilis back in 1996. There is no indication that these were removed prior to tree estimation.

Lines 365-372: The authors are over-interpreting their results: the suggestion that little HGT in H. monotropa means that mycoheterotrophs (MHT) are somehow indicative fundamentally different from parasitic plants in this regard should probably be tempered.

Lines 374-386: It would help readers to understand why RNA editing studies in mycoheterotrophs is important. There is no hypothesis presented in this manuscript for why it should be different from any other plant. Again, without comparing the edit sites to those known from other plants we cannot even assess whether they are unique in this species.

---

## Round 0.2 · Minor Revisions

· Academic Editor

Minor Revisions

Only one of the original reviewers agreed to handle the revised manuscript. That reviewer finds substantial problems with overinterpretation of the data and results. These should be addressed in the next revision.

Because of the limited number of reviewers, I took the time to review the manuscript myself. While I'm not an expert in this particular area, plant mitochondrial genomics, I know genomics pretty well. The analyses are well-suited to the study and the methods are well-done.

However, the manuscript itself is not particularly well-written and is in great need of editing by a native English speaker. It is difficult to interpret some sections and the lack of English language proficiency is distracting to the reader. I would recommend having a native English speaker revise the manuscript thoroughly while also addressing the comments or the reviewer.

Reviewer 3 ·

Basic reporting

The manuscript is largely clear and provides a nice data point from a well-known plant species but would benefit from minor editing for grammar.

The authors do a reasonable job summarizing the literature.

However, if they want to be more complete with regard to RNA-editing in parasitic plants on lines 457-458, where they state:

"By now the data on RNA-editing in mitochondria for non-photosynthetic plants are scarce. C-to-U RNA editing in seven genes (atp1, atp4, atp6, cox2, nad1, rps4, and rps12) was found in R. cantleyi (Xi et al., 2013)."

they could include the study by Barkman et al., 2007 which showed RNA editing in the putatively horizontally transferred atp1.

Experimental design

No problems for the most part. Individual gene trees are provided in SI.

Validity of the findings

The authors should be careful of over-interpreting results.

1. Lines 390-394: "All available evidence suggests that H. monotropa mitogenome has a linear fragment. While linear plasmids are found in mitochondrial genomes of several plants (e.g., in the work of Handa at el., (2002)), the linear fragment of the H. monotropa mitogenome lacks characteristic features of these plasmids: a terminal inverted repeat, the small size (not over 11 Kbp) and genes of RNA and DNA polymerases (reviewed in (Handa, 2008)).

While this may be true, the authors should be more cautious. The study would have benefited from use of more "direct" methods of analysis for genome structure. For instance, PCR or pulsed field gel electrophoresis (PFGE) is useful to document mt subgenomic circles, etc as has been done in other plants.


2. Lines 443-446: "MHT plants are fundamentally different in terms of the interaction with their hosts and, potentially, in many other features that stem from this. Thus the knowledge on MHT plant biology should be obtained using MHT models, not by the simple transfer of knowledge from the plant parasites."

I simply do not know of any scientists in the world that are trying to advance the notion that parasitic plants and mycoheterotrophs are some how identical and therefore whatever we know from parasitic plants must be true for MHT. Instead, I think the authors of this paper may be over-interpreting the results of their study in the opposite regard! Do the results from this study make the rule for all MHT?

3. Lines 470-473: "In many cases RNA editing has clear functional role (e.g. restoration of typical start codon in plastid genes rpl2 and psbL (Kudla et al., 1992). We found one such case in H. monotropa (nad4L). However, the dEN/dES value in mitochondrial transcripts is 1.23 (does not significantly differ from 1, p-value of 0.060) which indicates that in general edited sites are unlikely to have functional significance."

While I appreciate that the authors want to somehow quantify the proportion of edit events that either change or do not change the encoded amino acid for an ORF, to make the claim that since their metric is not different from 1 it means editing is not functionally important is erroneous. Even if only 1 out of 100 editing events changed a codon for an amino acid that altered a protein from non-functional to functional, wouldn't that be "functionally significant"? It may be true in this study that there are a lot of silent editing sites but to state that "in general edited sites are unlikely to have functional significance" is preposterous.

---

## Round 0.3 · Minor Revisions

· Academic Editor

Minor Revisions

This decision is based on comments by a section editor (Gerard Lazo) who makes the following comments and raises the associated concerns:

- The premise of the experimental observation is clear and it appears to be a valuable manuscript. The utility of the study is informative; however, has difficulty in actually viewing the sequence data of the main title topic. Though a sample sequence produced from another study is presented, the sequences assembled in the study were not provided; there were two mentioned. Following the reference given the complete genome was not readily viewable; is there a quick link to the accession or can it be recapitulated as a supplement? Issues were presented regarding difficulty in the assembly pipeline; however, it would be valuable to provide this as supplemental material. There are coordinate positions in supplemental table S2 and S3 for genes (and elsewhere); however, the source is not referenced in the legends. There is mention of SRA raw data; however, this is raw data and not the assembled products. There are numerous reference sequences with accessions for the comparative genomes; however, it is with great difficulty in following the current manuscript to get to the H. monotropa data presented in this manuscript. Additional clarity or attention is required.

- As I am asking for revisions to improve clarity; perhaps it would be valuable to add gene ontologies for the described genes as to generate GO: features pointing to the organelle origins and functional classification.

- Journal manuscripts are often scanned by text-mining software that locates and extracts core data elements, like gene function. Adding standard ontology terms, such as the Gene Ontology (GO, geneontology.org) or others from the OBO foundry (obofoundry.org) can enhance the recognition of your contribution and description. This will also make human curation of literature easier and more accurate. None of this was visible.

Organellar genomes have efficient pipelines for generation these days and we are of the opinion that a higher threshold is required to highlight the findings as a manuscript for publication. There were some reservations brought up for this manuscript; however, since this one leads us to such questioning we will outline some of the requirements. AFor the plastome study, this supports primarily request 2.iv. from our policies and procedures (https://peerj.com/about/policies-and-procedures/), but also has inklings of request 1. With the needed attention, I will additional request item 3 from https://peerj.com/about/policies-and-procedures/

Per https://peerj.com/about/policies-and-procedures/#discipline-standards subsection 4: Submissions reporting chloroplast genomes, mitochondrial genomes, etc. must

(1) report and compare three or more new complete sequences with phylogenetic implication with related groups
(2) For the one plastome sequence case, it should include at least one of following new finding(s) with its evolutionary implication:
(i) large gene content changes (losses, gains, or pseudogenizations);
(ii) large inversion(s) or translocation(s);
(iii) large IR expansion(s), contraction(s), loss(es), or shift(s);
(iv) other notable new evolutionary finding(s); or
(v) a phylogenetically new plastid sequence at order level or family level.
(3) if a gene list is included, appropriate annotations should be attached (e.g. GO:).

---

## Round 0.4 · accepted · Accept

· Academic Editor

Accept

Thank you for your responses.